# Decoupled Control for Double-T Dc-Dc MMC Topology for MT-HVdc/MVdc Grids

Cristián Pesce [1,*], Javier Riedemann [2], Rubén Peña [3], Iván Andrade [4], Werner Jara [5] and Rodrigo Villalobos [1]

1 Electrical Department, Universidad de La Frontera, Temuco 4780000, Chile
2 Electrical Department, University of Sheffield, Sheffield S10 2TN, UK
3 Electrical Department, Universidad de Concepción, Concepción 4030000, Chile
4 Electrical Department, Universidad de Magallanes, Punta Arenas 6200000, Chile
5 Electrical Department, Pontificia Universidad Católica de Valparaíso, Valparaíso 2340000, Chile
* Correspondence: cristian.pesce@ufrontera.cl

**Featured Application: The control method proposed in this work can be applied to power converters operating in multi-terminal high-voltage dc systems.**

**Abstract:** This paper proposes a decoupled control of a dc-dc modular multilevel converter (MMC) based on a double-T topology intended for multi-terminal high voltage direct current (MT-HVdc) transmission systems or emerging distribution systems operating in medium voltage direct current (MVdc). The aim of the proposed control strategy is to obtain an input current with reduced harmonic content and to eliminate the output ac common-mode voltage, which is not allowed in MT-HVdc systems. The control strategy consists of injecting two circulating ac currents and two dc currents that allow the energy balance between the arms of the converter and the general energy balance of the topology. The dc and ac currents are decoupled and allow control over load variations and reference changes in the dc-links. The proposed topology is mathematically modeled and the control method is then derived. Simulation results are presented to validate the proposed system.

**Keywords:** dc-dc power converters; modular multilevel converters; MT-HVdc converters; HVdc grids; MVdc grids

## 1. Introduction

A fundamental aspect of electric power systems is the efficiency of transmission lines from generation centers to consumption ones. This problem has led to important research and technological development in recent years [1–3]. Currently, most of the transmission grids in the world are based on high-voltage alternating current (HVac). However, HVac grids present serious limitations when it comes to transmitting at large distances [3–8], then power transmission in HVdc is the preferred alternative and has been largely implemented in different electrical systems around the world [9–12]. For instance, the ±1100 kV Ultra-HVdc line transmission in China [11] or the ±650 kV Kimal-Lo-Aguirre HVdc transmission line in Chile [13]. In the future, it is expected there will be an increase in the number of long-distance HVdc transmission systems [14–16]. On the other hand, an important configuration in HVdc grids is the multi-terminal connection grids (MT-HVdc) [17–19]. An MT-HVdc grid consists of the interconnection of three or more HVdc sub-stations with different voltage magnitudes (see Figure 1).

In MT-HVdc grids, there could be bidirectional power flows between the different terminals, therefore dc-dc converters are required to adapt the voltage levels of the individual substations, as shown in Figure 1 [20].

In recent years, research has been carried out on dc-dc converters for HVdc systems [21–24] and recently for MVdc grids [23]. Several power converter topologies for HVdc systems have been reported in the literature [25]. In particular, modular multilevel converters are

very attractive for this type of application [26,27], since they offer bidirectional power flow capability and low distortion in the input and output currents. Variations of the standard MMC structure have also been proposed, such as the resonant MMC [28–31], or the novel double-Y and double-Pi topologies [32–35].

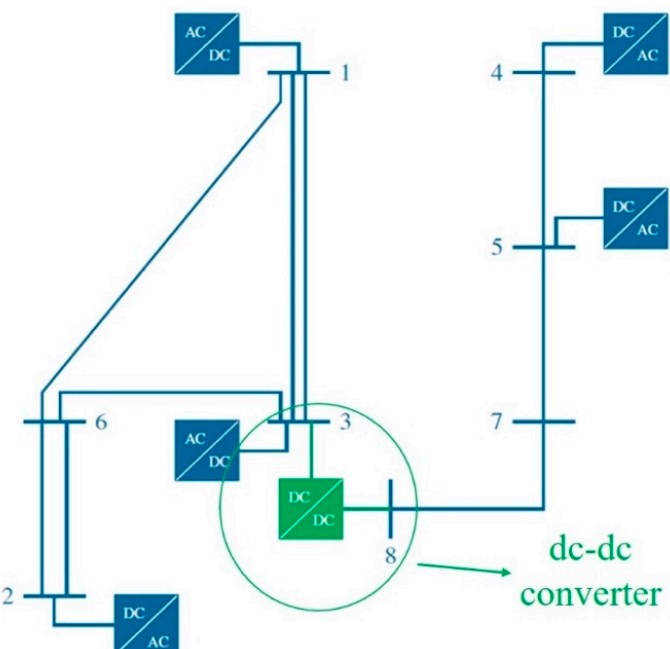

**Figure 1.** Schematic of a multi-terminal HVdc system.

In this paper, a decoupled control strategy for a dc-dc converter intended for MT-HVdc applications is presented. The topology has been called double-T since it is built with two T-structures (see Figure 2). The proposed MMC operating with the presented control method exhibits the following features: no common-mode voltage at the converter output, the minimum number of cells to build the topology, and null ac components in the input current. The operating principle of the converter is described in detail as well as the proposed control strategy. Simulation results are presented to validate the control method and a sensitivity analysis is carried out to evaluate its performance under variations of the converter parameters.

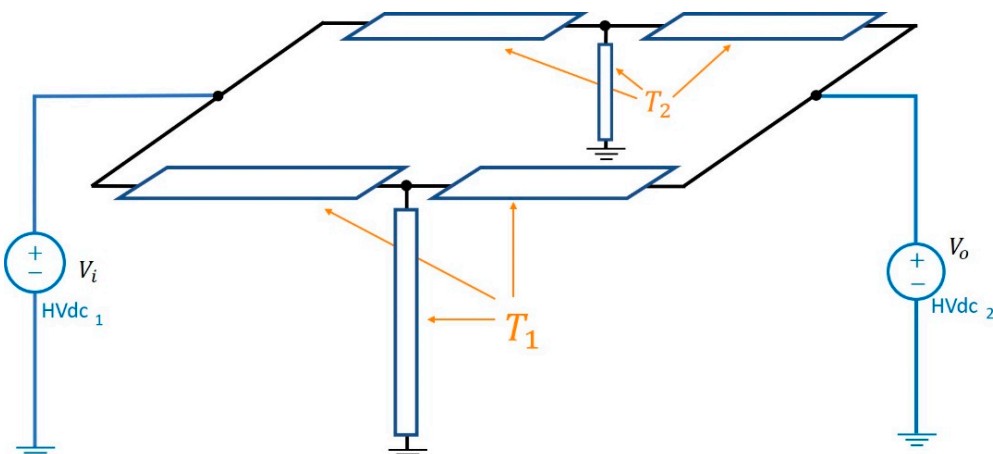

**Figure 2.** Double-T topology, with two T-structures called $T_1$ and $T_2$.

The remainder of the paper is organized as follows. In Section 2 the mathematical model of the converter is presented. Section 3 describes the operating principle of the

topology. The control strategy proposed is derived in Section 4 whereas the simulation results are presented in Section 5. In Section 6 Brief Sensitive Analysis is presented. Finally, the conclusions are stated in Section 7.

## 2. Converter Model

A schematic of the converter is shown in Figure 3. The converter is built with two T-structures ($T_1$ and $T_2$) with three arms each. In turn, each arm is built with an inductor $L$ and N cascaded H-bridges named *series*, *derivation*, and *output* stacks. The use of H-bridges in the stacks allows the possibility to operate with bidirectional power flow. Moreover, the inductors are required to couple the ac voltage components produced by each stack for current control purposes. The resistor $R$ represents the equivalent resistance of the inductor and each T-structure is grounded through a *derivation* stack.

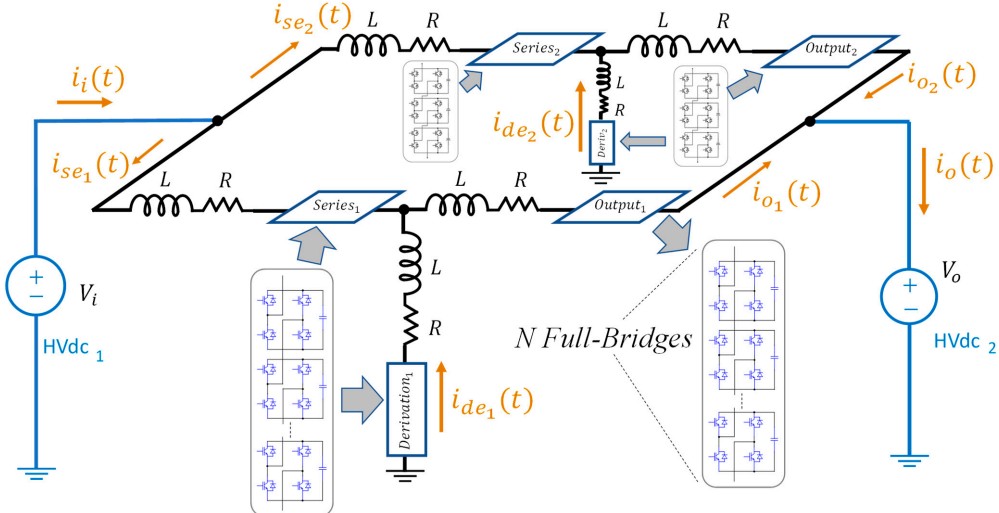

**Figure 3.** Scheme of the double-T topology.

Figure 4 shows the average model of the converter in terms of the voltages and currents in each arm of the topology. Time-varying voltages $v_{se_{1,2}}(t)$, $v_{de_{1,2}}(t)$, and $v_{o_{1,2}}(t)$ are produced by stacks *series*, *derivation*, and *output*, respectively. These time-varying voltages are dependent on the arm currents $i_{se_{1,2}}(t)$, $i_{de_{1,2}}(t)$, and $i_{o_{1,2}}(t)$, respectively. $V_i$ is the dc input voltage. $V_o$ is the converter output voltage which has a dc component and harmonics multiple of the switching frequency. In an MT-HVdc grid, $V_o$ is another terminal and can deliver power to $V_i$.

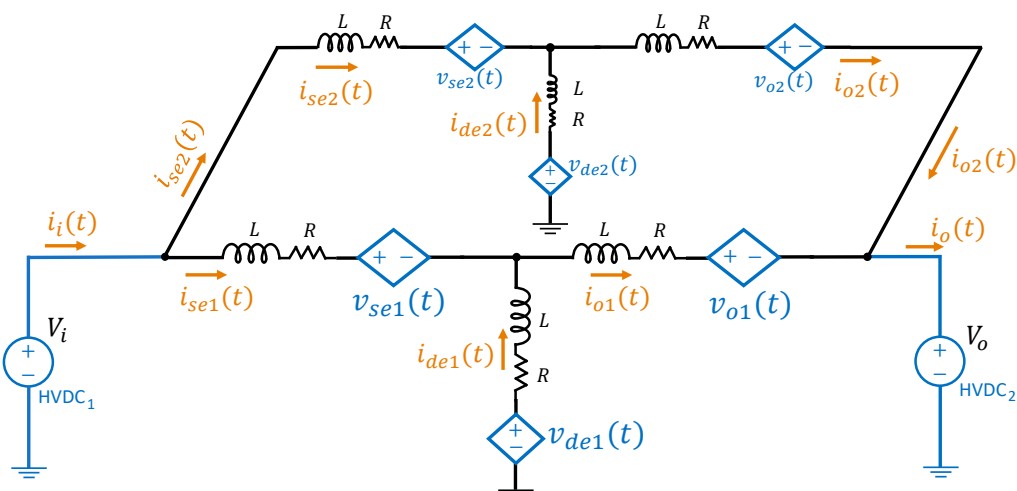

**Figure 4.** Average model of the double-T topology.

Applying voltage and current laws to the meshes, then the main equations defining the average model can be derived:

$$V_i = L\frac{di_{se1}(t)}{dt} - L\frac{di_{de1}(t)}{dt} + R(i_{se1}(t) - i_{de1}(t)) + v_{se1}(t) + v_{de1}(t) \tag{1}$$

$$V_i = L\frac{di_{se2}(t)}{dt} - L\frac{di_{de2}(t)}{dt} + R(i_{se2}(t) - i_{de2}(t)) + v_{se2}(t) + v_{de2}(t) \tag{2}$$

$$0 = L\frac{di_{de1}(t)}{dt} + L\frac{di_{o1}(t)}{dt} + R(i_{de1}(t) + i_{o1}(t)) - v_{de1}(t) + v_{o1}(t) + V_o \tag{3}$$

$$0 = L\frac{di_{de2}(t)}{dt} + L\frac{di_{o2}(t)}{dt} + R(i_{de2}(t) + i_{o2}(t)) - v_{de2}(t) + v_{o2}(t) + V_o \tag{4}$$

$$i_{de1}(t) = i_{o1}(t) - i_{se1}(t) \tag{5}$$

$$i_{de2}(t) = i_{o2}(t) - i_{se2}(t) \tag{6}$$

$$-i_i(t) - i_{de1}(t) - i_{de2}(t) + i_o(t) = 0 \tag{7}$$

The voltages $v_{se_{1,2}}(t)$, $v_{de_{1,2}}(t)$ and $v_{o_{1,2}}(t)$ are defined in (8)–(10) where $S_{i,se_{1,2}}(t)$, $S_{i,de_{1,2}}(t)$ and $S_{i,o_{1,2}}(t)$ are the switching functions of each sub-module (H-bridge) of each stack of the topology. The value of the switching functions can be either $-1$, $0$, or $1$. Finally, $C_i$ are the capacitors of each sub-module.

$$v_{se_{1,2}}(t) = \sum_{j=1}^{2}\sum_{i=1}^{N}\frac{1}{C_i}\int\left(i_{se_j}(t)\cdot S_{i,se_j}(t)\right)dt \tag{8}$$

$$v_{de_{1,2}}(t) = \sum_{j=1}^{2}\sum_{i=1}^{N}\frac{1}{C_i}\int\left(i_{de_j}(t)\cdot S_{i,de_j}(t)\right)dt \tag{9}$$

$$v_{o_{1,2}}(t) = \sum_{j=1}^{2}\sum_{i=1}^{N}\frac{1}{C_i}\int\left(i_{o_j}(t)\cdot S_{i,o_j}(t)\right)dt \tag{10}$$

From (1)–(7) it is possible to obtain the expressions for the instantaneous currents. Let $x(t) = [i_{o1}(t)\ i_{o2}(t)\ i_{se1}(t)\ i_{se2}(t)]^T$ be the state vector of independent currents and $u(t) = [v_{se1}(t)\ v_{se2}(t)\ v_{de1}(t)\ v_{de2}(t)\ v_{o1}(t)\ v_{o2}(t)]^T$ the control variables vector, the state-space model is given by:

$$\frac{d}{dt}\begin{bmatrix}i_{o1}(t)\\i_{o2}(t)\\i_{se1}(t)\\i_{se2}(t)\end{bmatrix} = -\frac{R}{L}\begin{bmatrix}i_{o1}(t)\\i_{o2}(t)\\i_{se1}(t)\\i_{se2}(t)\end{bmatrix} + \frac{1}{3L}\begin{bmatrix}-1 & 0 & 1 & 0 & -2 & 0\\0 & -1 & 0 & 1 & 0 & -2\\-2 & 0 & -1 & 0 & -1 & 0\\0 & -2 & 0 & -1 & 0 & -1\end{bmatrix}\begin{bmatrix}v_{se1}(t)\\v_{se2}(t)\\v_{de1}(t)\\v_{de2}(t)\\v_{o1}(t)\\v_{o2}(t)\end{bmatrix} + \begin{bmatrix}V_i - 2V_o\\V_i - 2V_o\\2V_i - V_o\\2V_i - V_o\end{bmatrix} \tag{11}$$

## 3. Operating Principle

This work considers an approach based on the energy balance between the stacks of the converter. To achieve this balance ac and dc voltages and currents are employed to transfer energy to and from the different stacks. Figure 5 shows a simplified model using dc voltages $V_{se_{1,2}}$, $V_{de_{1,2}}$, $V_{o_{1,2}}$ and ac voltages $v_{se_{1,2}}$, $v_{de_{1,2}}$, $v_{o_{1,2}}$ that are produced by each stack, under steady-state operation. The dc voltages are obtained by applying voltage law to the arms of the topology; their values are $V_{se_{1,2}} = (V_i - V_o)$ and $V_{de_{1,2}} = V_o$. For control purposes the dc voltages of the *output* stacks are zero. If $V_{o_{1,2}} = 0$, the energy transferred by the *output* stacks due to dc components is minimized, and it is associated mainly to ac voltages and currents.

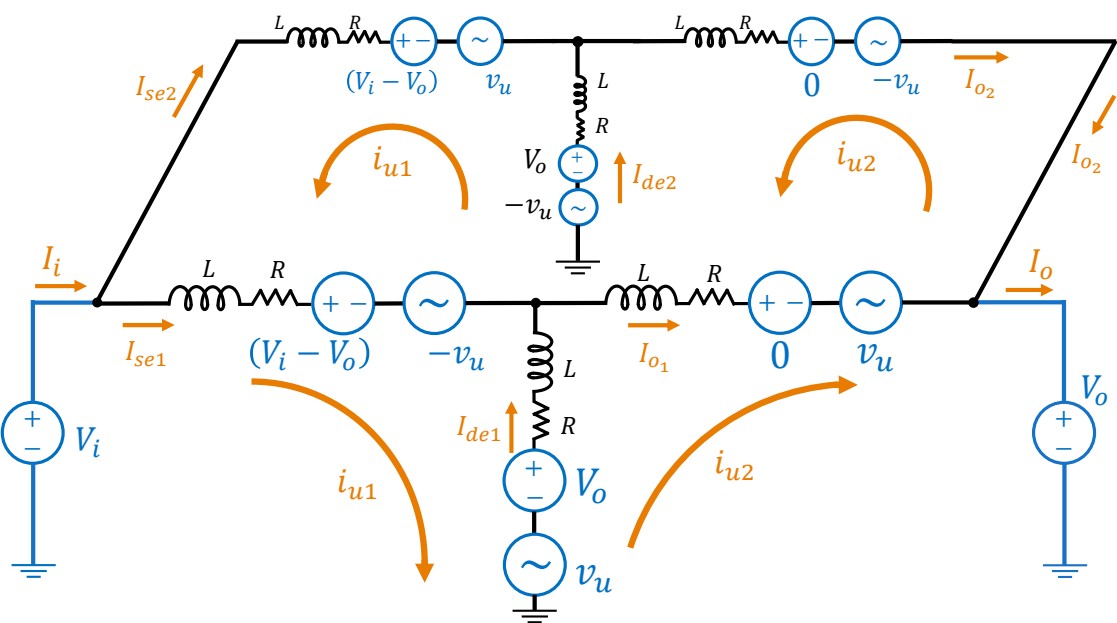

**Figure 5.** Simplified scheme of the topology with ac and dc voltages and circulating currents.

The ac voltages $v_{se1,2}$, $v_{de_{1,2}}$, $v_{o_{1,2}}$ have sinusoidal waveform as $\pm v_u = \pm V_u \sin(2\pi f_c t)$, where $f_c$ is a fundamental frequency. The sign of $V_u$ depends on the control objective and on the circulating current, in this case $v_{se1} = -v_u$, $v_{se_2} = v_u$, $v_{de_1} = +v_u$, $v_{de_2} = +v_u$, $v_{o_1} = +v_u$ and $v_{o_2} = -v_u$ (see Figure 5). These ac voltages are intended to interact with the circulating ac currents $i_{u1}$ and $i_{u2}$ to achieve the energy balance of the converter.

The ac circulating currents can be expressed as $i_{u1} = I_{u1} \sin(2\pi f_c t)$ and $i_{u2} = I_{u2} \sin(2\pi f_c t)$. These currents are aimed to allow the energy balance of the *derivation* ($i_{u1}$) and *output* ($i_{u2}$) stacks. Additionally, dc currents $I_{se1,2}$, $I_{de1,2}$, $I_{o1,2}$ circulate in all arms of the topology. The dc currents $I_{se1,2}$ allow the energy balance of *series* stacks. Using expressions (1)–(7), neglecting the series equivalent resistance $R$ of the inductor $L$ and neglecting the product $\omega L$ which is comparatively small with respect to the product $VI$, it is possible to calculate the power equations of each stack as defined in (12)–(17). With the operating principle of the topology and the model defined in Section 2, the control strategy can now be derived as presented in Section 4.

$$P_{se_1} = \frac{d\overline{W}_{se_1}}{dt} = \int v_{se1}(t) \cdot i_{se1}(t) dt = (V_i - V_o) I_{se_1} - v_u \, i_{u1} \tag{12}$$

$$P_{de_1} = \frac{d\overline{W}_{de_1}}{dt} = \int v_{de1}(t) \cdot i_{de1}(t) dt = -V_o I_{de_1} + v_u \, (i_{u1} - i_{u2}) \tag{13}$$

$$P_{o_1} = \frac{d\overline{W}_{o_1}}{dt} = \int v_{01}(t) \cdot i_{o1}(t) dt = v_u \, i_{u2} \tag{14}$$

$$P_{se_2} = \frac{d\overline{W}_{se_2}}{dt} = \int v_{se2}(t) \cdot i_{se2}(t) dt = (V_i - V_o) I_{se_2} - v_u \, i_{u1} \tag{15}$$

$$P_{de_2} = \frac{d\overline{W}_{de_2}}{dt} = \int v_{de2}(t) \cdot i_{de2}(t) dt = -V_o I_{de_2} - v_u \, (-i_{u1} + i_{u2}) \tag{16}$$

$$P_{o_2} = \frac{d\overline{W}_{o_2}}{dt} = \int v_{o2}(t) \cdot i_{o2}(t) dt = v_u \, i_{u2} \tag{17}$$

## 4. Control of the Double-T Converter

The control strategy is based on the average energy balance of all arms of the converter. The control scheme is composed by three stages: average energy control on each stack;

current control with references obtained in the energy control; and stack internal control for the equalization of the dc-link voltage in each submodule.

*A. Energy Balance of Stacks*

Using expressions (1)–(4) and (11) and knowing that the independent currents of the topology are four, the control system is designed. Taking into account the energy balance approach, the energy of the *series* stack is controlled by $I_{se}$ (see (12) and (15)) via a proportional–integral (*PI*) controller. For *derivation* stacks, the control action is performed by the ac current $i_{u1}$ whereas for *output* stacks, the energy control is carried out by the ac current $i_{u2}$. It should be noted in Figure 5 that $i_{u1}$ and $i_{u2}$ circulate through both T-structures. Therefore, for the energy control of the converter, there are four *PI* controllers.

To impose the dc currents $I_{se_1}$ and $I_{se_2}$ from energy *series*$_{1,2}$ controllers, another two *PI* controllers are used. As the energy control loops and the current control loops are nested, in this work they are decoupled using natural frequencies $\omega_n$ of the controllers in a ratio of 1/70. The natural frequency of the *series*$_{1,2}$ energy controllers is $\omega_{n_e}$ and the natural frequency of the current controllers is $\omega_{n_c}$, then $\omega_{n_c}/\omega_{n_e} = 70$. Figure 6 shows the control scheme for the energy balance of the *series* stacks.

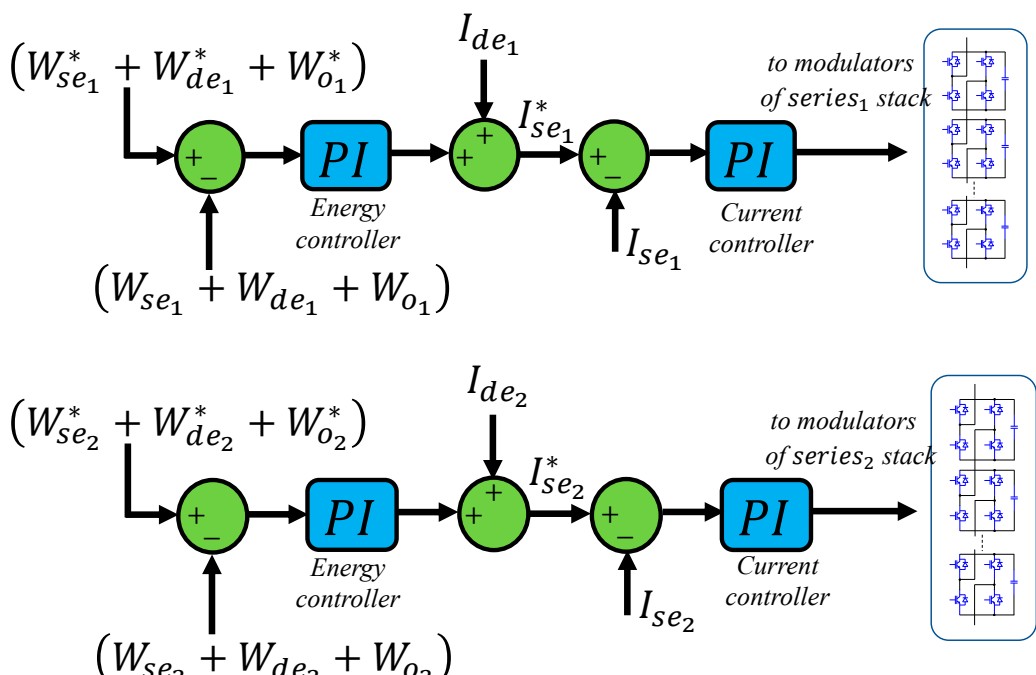

**Figure 6.** *Series* stacks energy and current control scheme.

To impose alternating currents $i_{u1}$ and $i_{u2}$ from energy controllers in *derivation* stacks, proportional–resonant (*PR*) controllers are used. In this work, the frequency used for $i_{u1}$ and $i_{u2}$ is $f_c = 500$ [Hz]. This frequency should consider the "trade-off" between switching losses in the H-bridges and the dynamic response in the *derivation* energy balance controllers. Additionally, in this work, for the proposed operating point, the magnitude of $v_u$ is constant as $V_u = 0.7V_o$. The value of $V_u$ is fundamental for the energy control of the *derivation* and *output* stacks. It is desired for $V_u$ to be high so that the dynamics of the control are fast, but it must not exceed the maximum output voltage to avoid overmodulation and the consequent voltage distortion. For the simulation study $V_u = 9.1$ kV. Since the output direct voltage is $V_o = 13kV_{DC}$, then $V_{arm} = 13$ kV $+ 9.1$ kV $= 22.1$ kV is within the value that the converter arm can deliver. The maximum value to be delivered by the arm is $V_{arm_{max}} = 3$ kV·8 $= 24$ kV. Figure 7 shows the energy balance control scheme of the converter stacks.

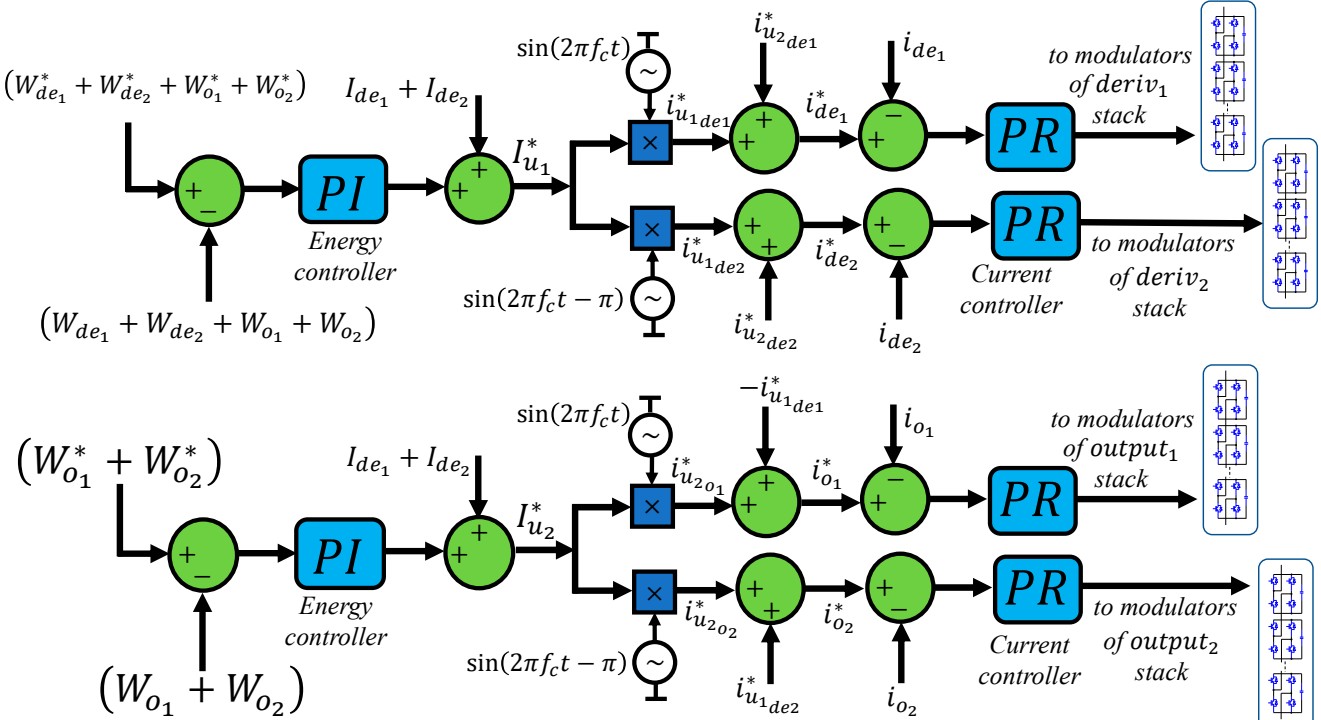

**Figure 7.** *Derivation* and *output* stacks energy and current control scheme.

*B.* *Intra-Stacks Voltage Balance*

The balance of the dc-links voltages of each stack is carried out by exchanging power between a quadrature current $i_q$ that circulates in addition to the energy balance currents. This current can be defined as $i_q = I_q \sin(\omega_c t + \pi/2)$ and is phase-shifted 90° with respect to $i_{u1}$ and $i_{u2}$. Then, $i_{u1} + i_q$ and $i_{u2} + i_q$ are the currents circulating internally in the converter. Moreover, in each sub-module there is a voltage $v_{bal} = V_{bal} \sin(\omega_c t + \pi/2)$ that is in phase with the current $i_q$. This allows exchanging power and generating the internal balance of the stacks keeping the dc-links of the N sub-modules with the same voltage. This voltage balance controller is based on a proportional *(P)* controller that receives the average value of the N dc-links voltages. The controller output is multiplied by $\sin(\omega_c t + \pi/2)$ and $V_{bal_N}$ is then generated. Finally, as $i_q$ is in quadrature with respect to the currents $i_{u1}$ and $i_{u2}$, the intra-stack balance acts decoupled with respect to the energy control.

*C.* *Decoupling Energy Balance Control*

According to Equations (12)–(14), if these expressions are added, then (18) is obtained. In the same way, if (15)–(17) are added, the expression (19) is obtained.

$$P_{se_1} + P_{de_1} + P_{o_1} = (V_i - V_o)I_{se_1} - V_o I_{de_1} \tag{18}$$

$$P_{se_2} + P_{de_2} + P_{o_2} = v_u\, i_{u2} \tag{19}$$

Equations (18) and (19) are used to design the *series* stack energy controllers. The dc currents $I_{de_{1,2}}$ are added to obtain a feedforward compensation (see Figure 7). This leads to an improved energy balance dynamic of the dc-links of the *series* stacks. In the same way for the design of the *derivation* stacks energy control loops, if the (13), (14), (16), and (17) are added, the result is (20). In Figure 7 the implementation of the control strategy can be seen. For the control design of output stacks energy loops (*output*₁ and *output*₂), as there is no power transfer of DC components, this decoupling principle does not apply.

$$P_{de_1} + P_{o_1} + P_{de_2} + P_{o_2} = 2v_u i_{u_1} - V_o(I_{de_1} + I_{de_2}) \tag{20}$$

## 5. Simulations Results

Simulations are carried out considering the parameters in Table 1. Since 8 sub-modules per stack are used, the topology comprises a total of 48 H-bridges. The input and output voltage values have been defined for a typical MVdc distribution system.

**Table 1.** Simulation parameters.

| Description | Value |
|---|---|
| Input voltage, $HVDC_1 = V_{in}$ | 23000 [V] |
| Output voltage, $HVDC_2 = V_o$ | 13000 [V] |
| Rated output power, $P_{out}$ | 1.25 [MW] |
| Dc-links capacitors, $C_{se_{1,2}} = C_{de_{1,2}} = C_{o_{1,2}}$ | 1 [mF] |
| Arm inductances, $L$ | 2.5 [mH] |
| Series equivalent resistance of inductors, $R$ | 50 [mΩ] |
| Dc-links reference voltage | 3000 [V] |
| Frequency of ac circulating currents | 500 [Hz] |
| Number of H-bridges per stack | 8 |
| Number of arms | 6 |
| Switching frequency, $f_{sw}$ | 2 [kHz] |

Since the focus of this paper is on the control strategy for the double-T topology, the design aspects of the converter are not addressed in detail. However, general design procedures reported in [32,34,35] were followed.

Regarding the nomenclature used to show the results, the capacitor of the first H-bridge of the *series*$_{1,2}$, *derivation*$_{1,2}$, and *output*$_{1,2}$ stacks will be called $v_{c1_{se1,2}}$, $v_{c1_{de1,2}}$ and $v_{c1_{o1,2}}$, respectively.

In the first test, simulations for load changes applied at t = 0.4 s, t = 0.8 s, and t = 1.2, are performed. Figure 8a shows one dc-link voltage per stack. As can be seen, the energy balance controller responds satisfactorily and very small variations in the dc-link voltages are obtained when the load changes. The voltage reference for all the dc-links has been set to 3 kV. Figure 8b shows the output power. From 0.8 to 1.2 s the output power is nominal.

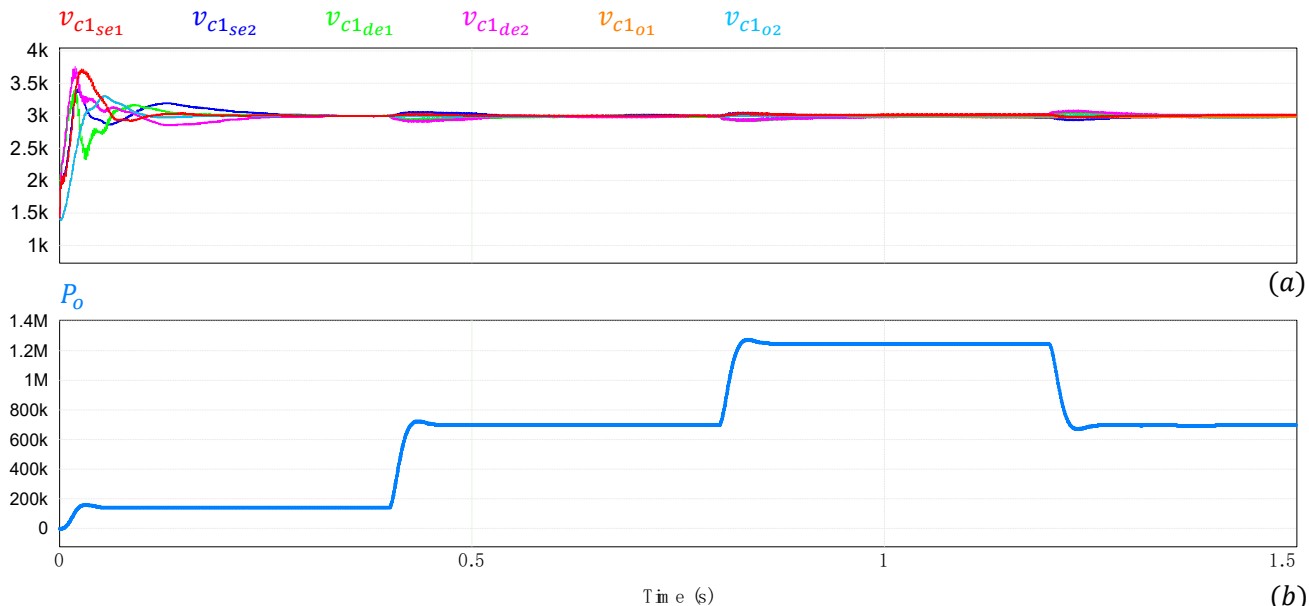

**Figure 8.** (**a**) One dc-link voltage per stack. (**b**) Converter output power.

Figure 9 shows the 24 voltage waveforms of the dc-links of the $T_1$ structure whereas Figure 10 shows the dc-link voltages associated with the $T_2$ structure. An accurate response

of the control scheme can be appreciated, with a minimum variation on the dc-link voltages when the load varies.

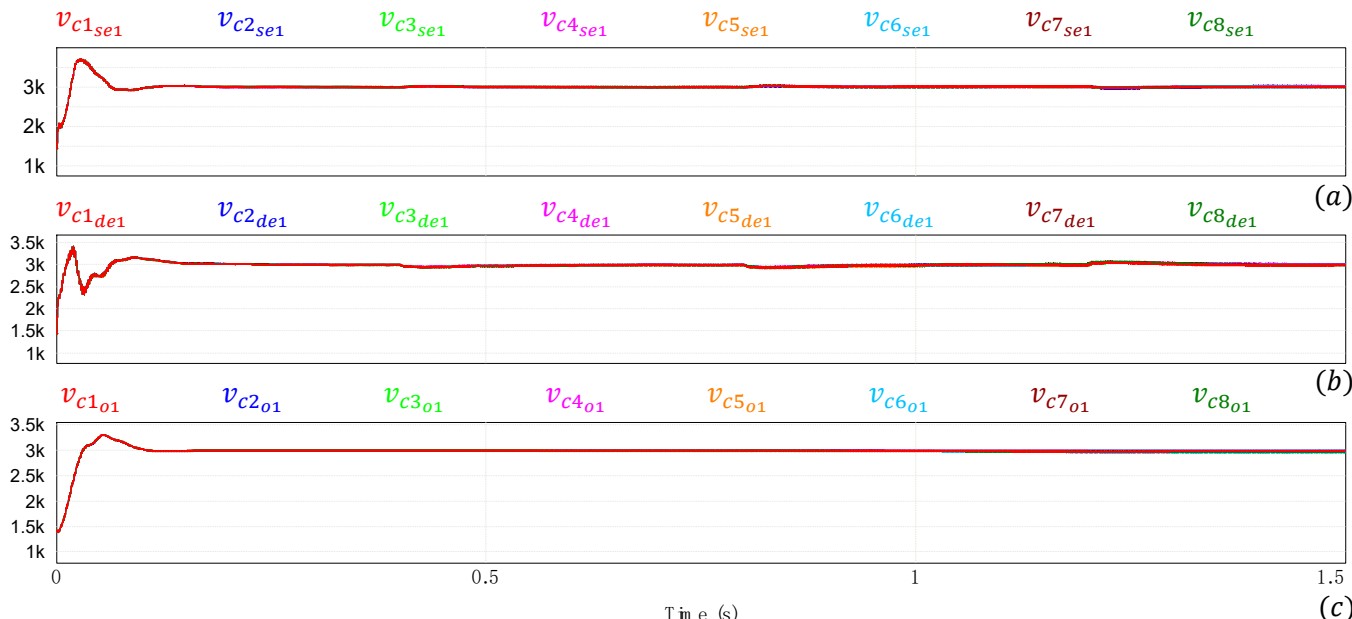

**Figure 9.** $T_1$ dc-link voltages. (**a**) *series*$_1$ stack. (**b**) *derivation*$_1$ stack. (**c**) *output*$_1$ stack.

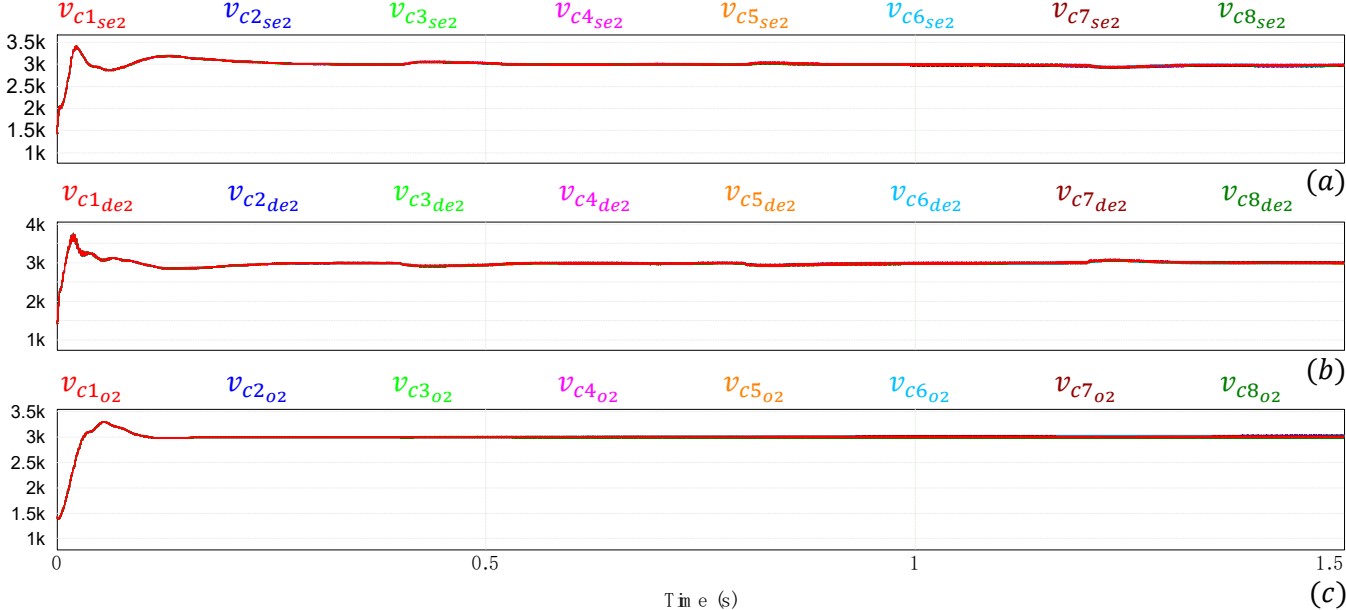

**Figure 10.** $T_2$ dc-link voltages. (**a**) *series*$_2$ stack. (**b**) *derivation*$_2$ stack. (**c**) *output*$_2$ stack.

The converter output voltage and input current are shown in Figure 11, both presenting very low ripple. The spectra associated to these waveforms are shown in Figure 12. In both waveforms, the THD is lower than 1% with respect to the fundamental frequency, which in this case is 0 Hz. The average voltage value is $V_{0_{0Hz}} = 13$ kV and the average current is $I_{in_{0Hz}} = 53$ A. As the number of sub-modules per stack are 8 and the switching frequency is 2 kHz, the harmonic components appear around 32 kHz (since the H-bridge doubles the equivalent switching frequency).

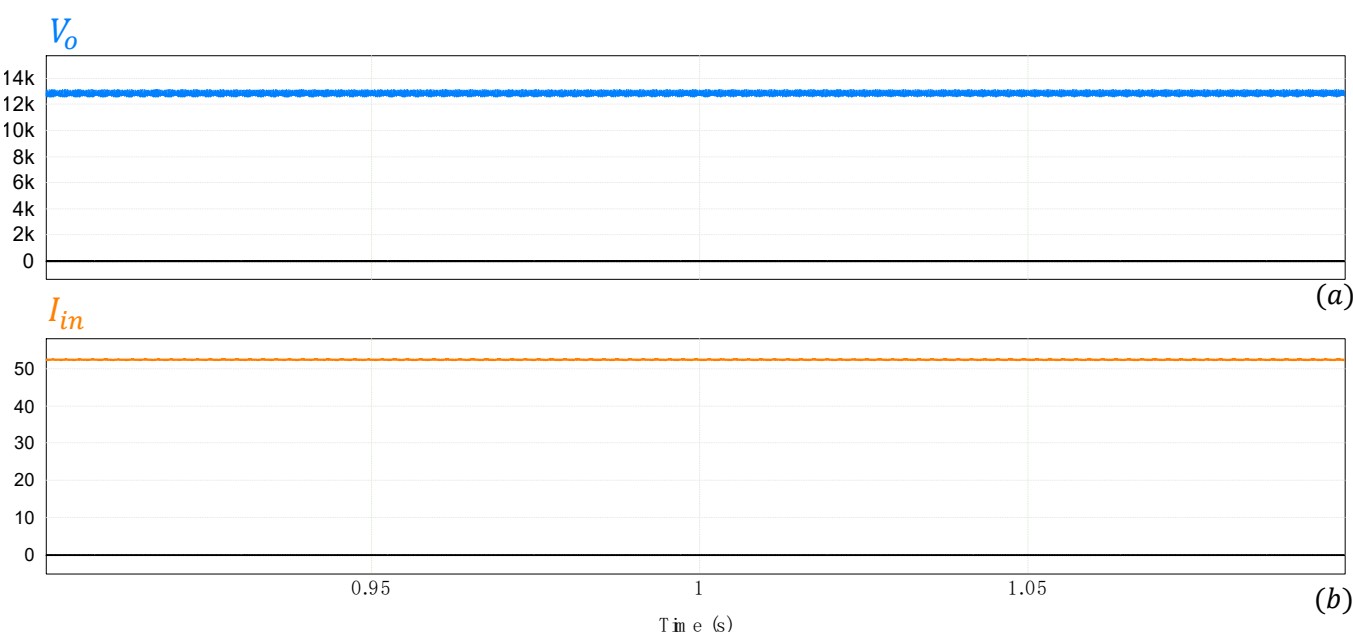

**Figure 11.** (**a**) Output voltage. (**b**) Input current.

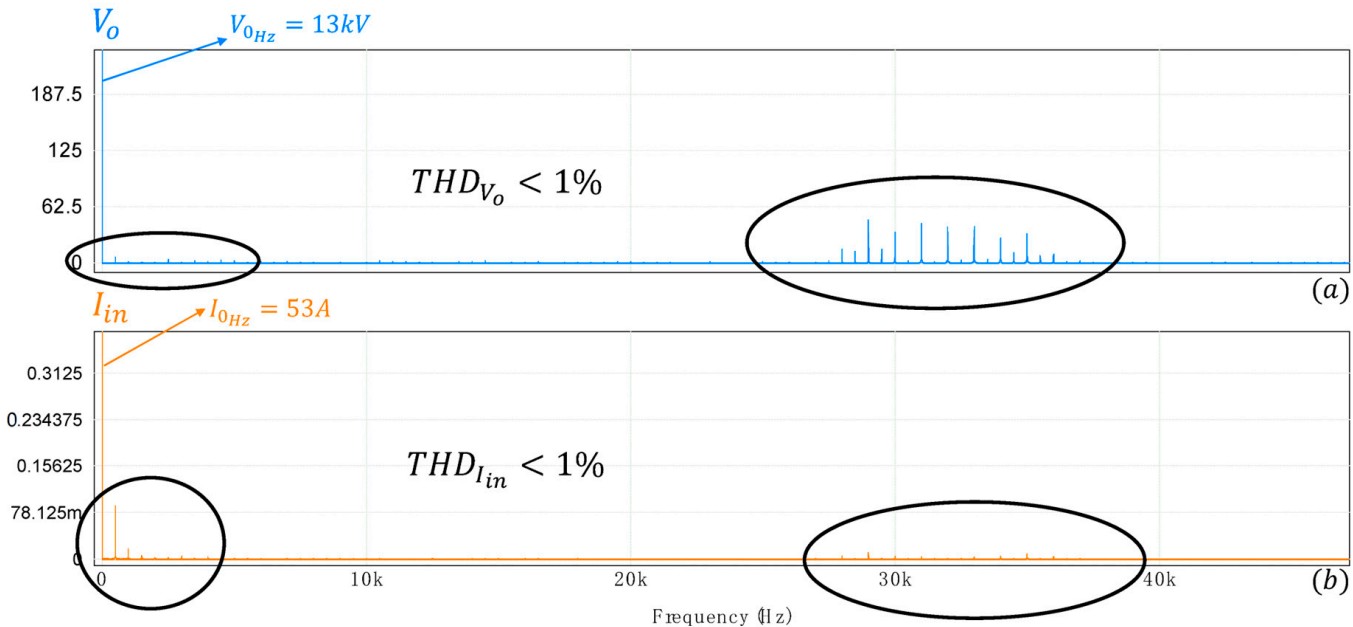

**Figure 12.** (**a**) Output voltage spectra. (**b**) Input current spectra.

Figure 13 shows the currents of each converter arm. The currents in the *series* and *derivation* stacks are shown in Figure 13a,b, respectively. It can be noted that these currents have both ac and dc components. On the other hand, the current in the *output* stacks in Figure 13c is mainly dc with a small ac component. It is important to note that the currents $i_{se1}$ and $i_{se2}$ must be phase-shifted 180° so that the converter input current $I_{in}$ has negligible ac component (see Figure 11b).

Figure 14 shows results for power variation and changes in the dc-link voltage reference. One dc-link voltage per stack is shown in Figure 14a and the output power is presented in Figure 14b. At t = 0.5 s a change in the reference of the dc-link voltages from 3 kV to 3.6 kV is applied. The opposite change is applied at t = 0.8 s. Good dynamic response of the dc-link voltages in every stack can be observed.

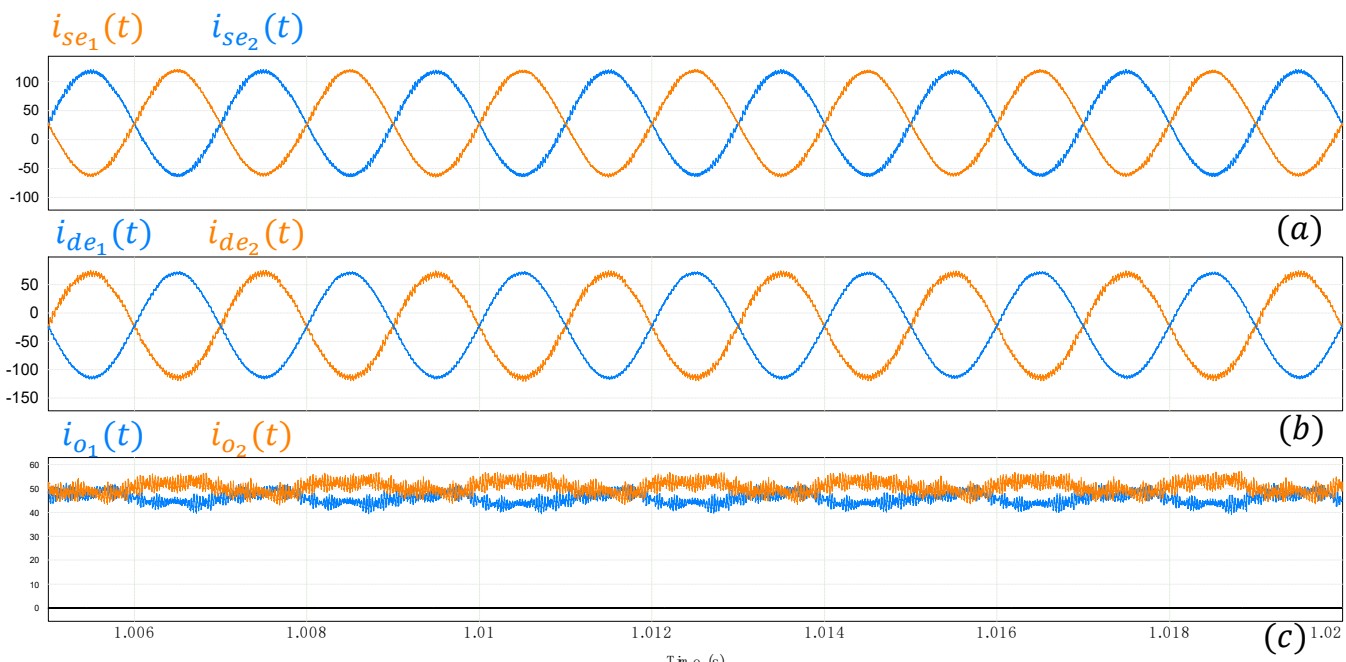

**Figure 13.** (**a**) $i_{se_1}(t)$ and $i_{se_2}(t)$ currents. (**b**) $i_{de_1}(t)$ and $i_{de_2}(t)$ currents. (**c**) $i_{o_1}(t)$ and $i_{o_2}(t)$ currents.

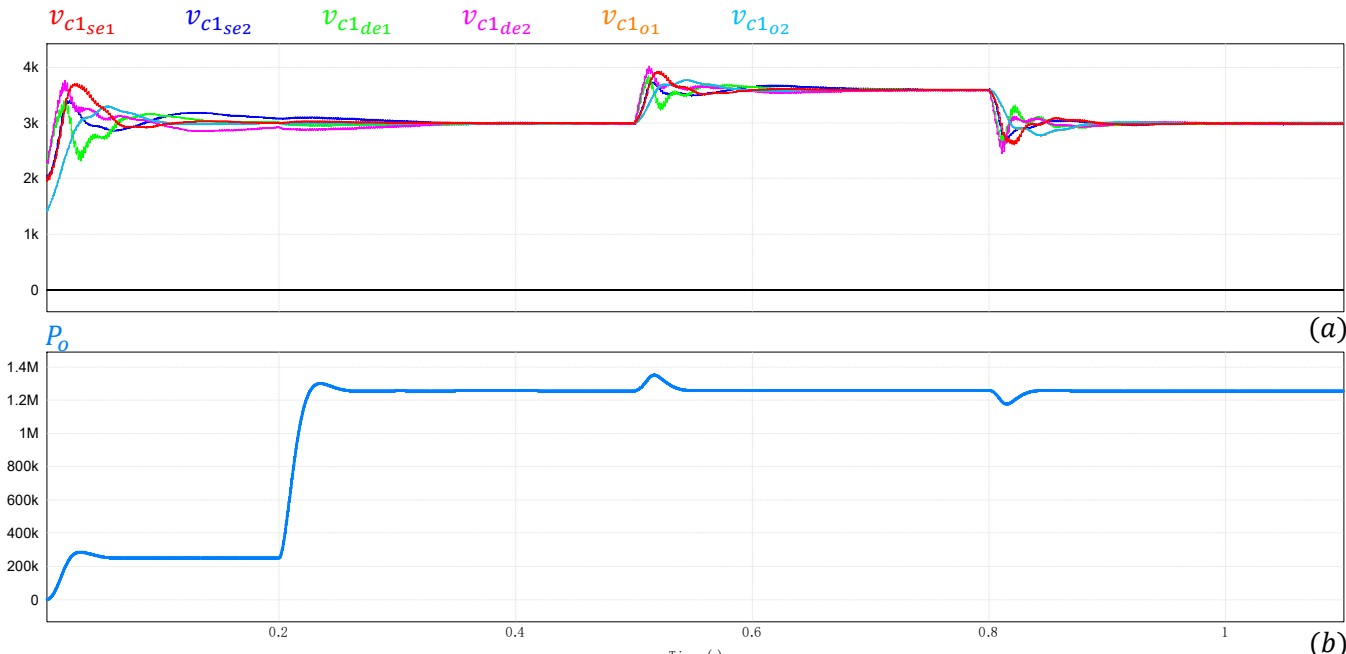

**Figure 14.** (**a**) One dc-link voltage per stack. (**b**) Converter output power (for for dc-links voltage control reference changes).

Figure 15 shows results for variations in the input and output voltages. In the case of the output voltage, the control reference is varied. Figure 15a shows the waveforms of one dc-link voltage per stack. Changes in the output voltage reference take place at t = 0.4 s and t = 0.8 s, whereas a change in the input voltage is applied at t = 0.6 s. In every input/output voltage variation applied, the fast and accurate response of the dc-link voltage control system can be observed.

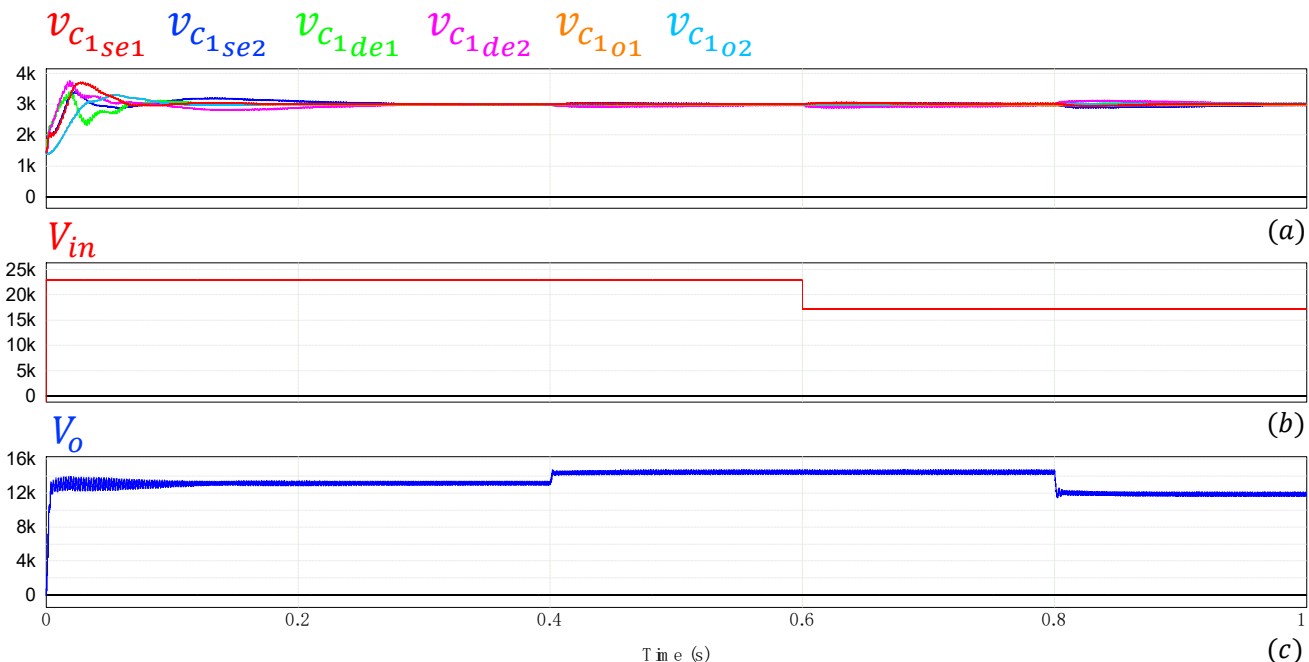

**Figure 15.** (**a**) One dc-link voltage per stack. (**b**) Converter input voltage (**c**) Converter output voltage.

## 6. Brief Sensitivity Analysis

A study to validate the performance of the control system under changes in parameters of the converters is carried out. Variations of the capacitances and inductances of the converters are considered, and the parameters of the controllers are not changed.

### 6.1. Case 1

The value of the capacitors are set to $C_{se_1} = 1$ mF, $C_{se_2} = 0.99$ mF, $C_{de_1} = 1$ mF, $C_{de_2} = 1.01$ mF, $C_{o_1} = 1$ mF, and $C_{o_2} = 1.01$ mF. It should be mentioned that all the capacitors in a single arm are equal. The dc-link voltage reference is 3 kV and the converter operates with nominal load.

Figure 16 shows the dc-link voltages obtained. As can be seen, the control response is still accurate reaching the reference in short time and without error in steady state.

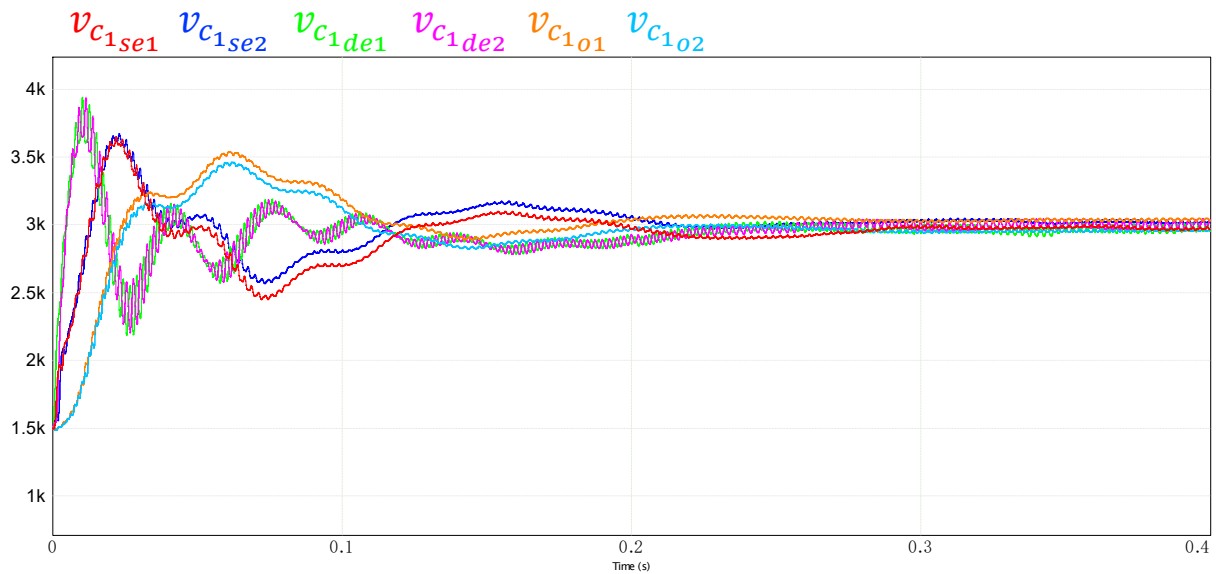

**Figure 16.** Dc-link voltage waveforms for case 1.

### 6.2. Case 2

In this second analysis, the values of the capacitors are: $C_{se_1} = 0.98$ mF, $C_{se_2} = 1.02$ mF, $C_{de_1} = 1$ mF, $C_{de_2} = 1.02$ mF, $C_{o_1} = 0.98$ mF, and $C_{o_2} = 1.02$ mF. The results are shown in Figure 17. Again, a good response of the control system and zero steady-state error are obtained.

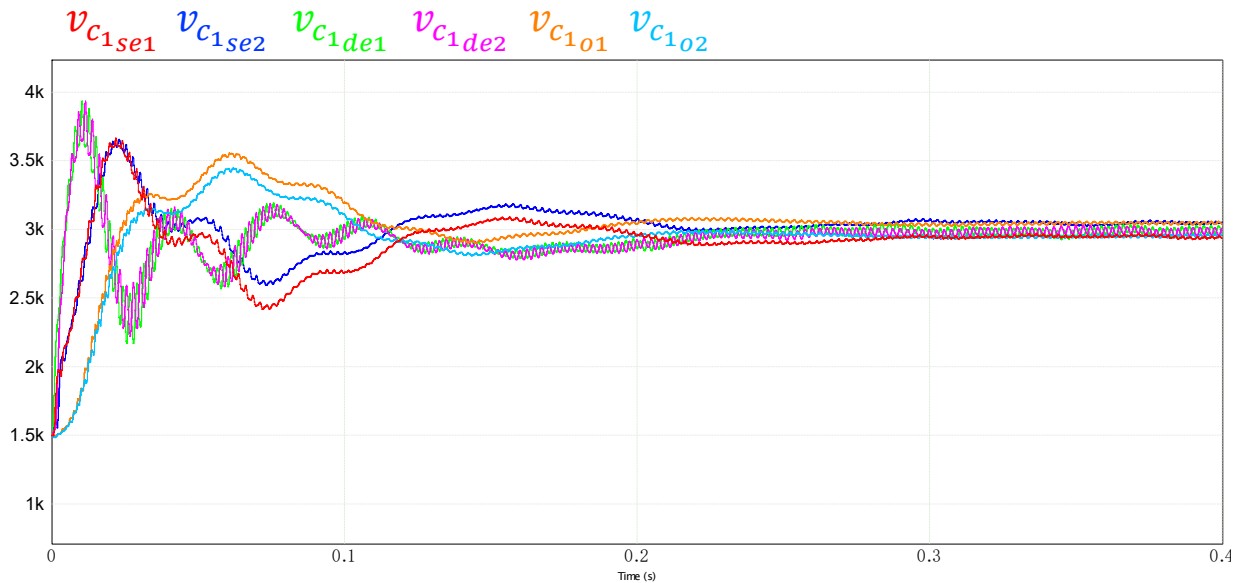

**Figure 17.** Dc-link voltage waveforms for case 2.

### 6.3. Case 3

In this third analysis, the capacitor values are: $C_{se_1} = 0.9$ mF, $C_{se_2} = 1.1$ mF, $C_{de_1} = 0.95$ mF, $C_{de_2} = 1.05$ mF, $C_{o_1} = 0.9$ mF, and $C_{o_2} = 1$ mF. The dc-link voltage waveforms obtained are shown in Figure 18. In this case, the variation of the parameters has been in some arms more than 10% of the original value and certain steady-state errors can now be observed.

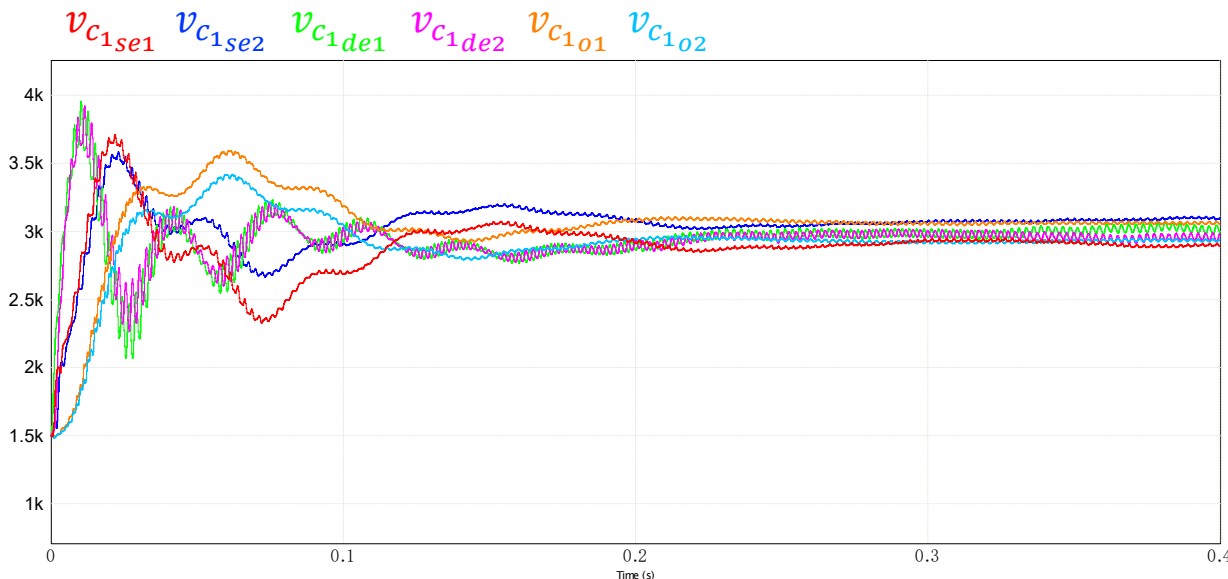

**Figure 18.** Dc-link voltage waveforms for case 3.

In the following cases 4, 5, and 6, the effect on the control system due to variations in the inductances is analyzed.

### 6.4. Case 4

In this first inductance variation scenario, the values of the inductors are set to $L_{se_1} = 2.51$ mH, $L_{se_2} = 2.5$ mH, $L_{de_1} = 2.49$ mH, $L_{de_2} = 2.51$ mH, $L_{o_1} = 2.49$ mH, and $L_{o_2} = 2.5$ mH. The dc-link voltages are presented in Figure 19 where a proper dynamic response and null steady-state error are appreciated.

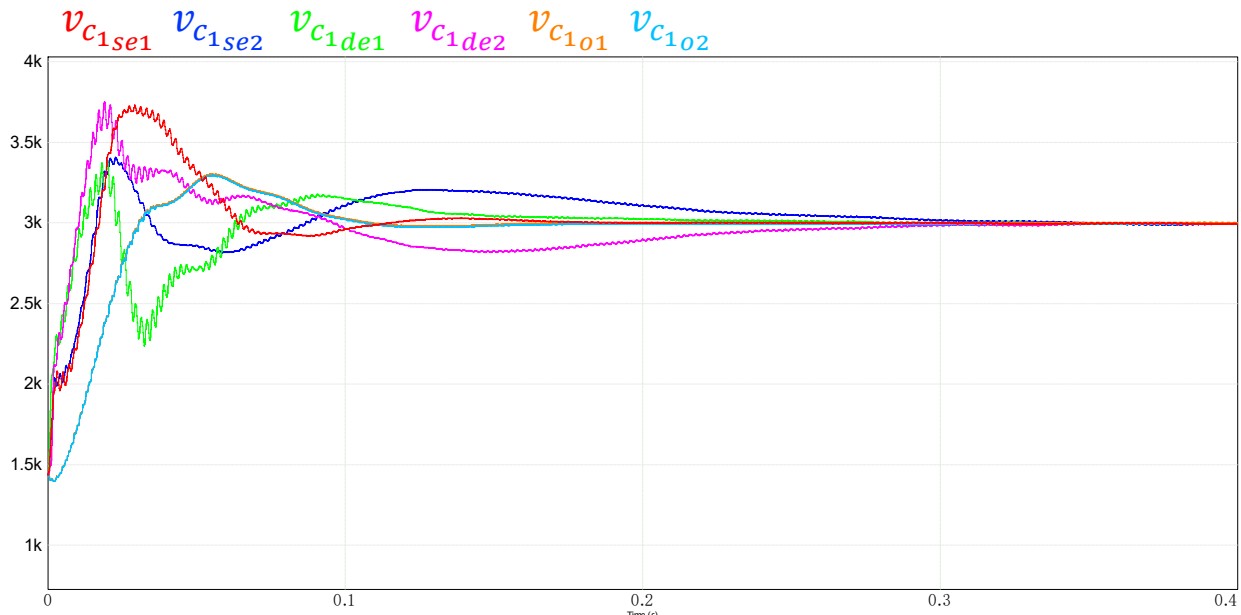

**Figure 19.** Dc-link voltage waveforms for case 4.

### 6.5. Case 5

The values of the inductors are: $L_{se_1} = 2.45$ mH, $L_{se_2} = 2.55$ mH, $L_{de_1} = 2.46$ mH, $L_{de_2} = 2.54$ mH, $L_{o_1} = 2.47$ mH, and $L_{o_2} = 2.5$ mH, and the dc-link voltages are presented in Figure 20. Once again, the control responds properly and there is no error in the steady state.

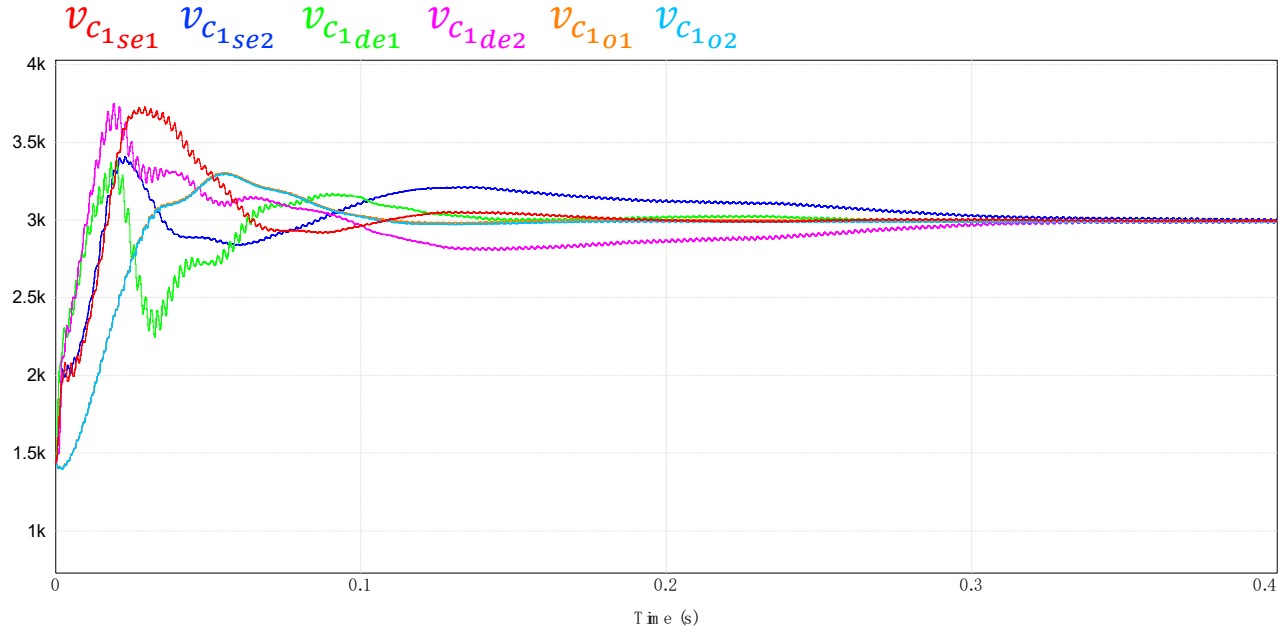

**Figure 20.** Dc-link voltage waveforms for case 5.

### 6.6. Case 6

Finally, the inductor values are defined as: $L_{se_1} = 2.3\,\text{mH}$, $L_{se_2} = 2.65\,\text{mH}$, $L_{de_1} = 2.4\,\text{mH}$, $L_{de_2} = 2.45\,\text{mH}$, $L_{o_1} = 2.6\,\text{mH}$, and $L_{o_2} = 2.7\,\text{mH}$. For this condition, the results are shown in Figure 21 where a response similar to cases 4 and 5 is obtained.

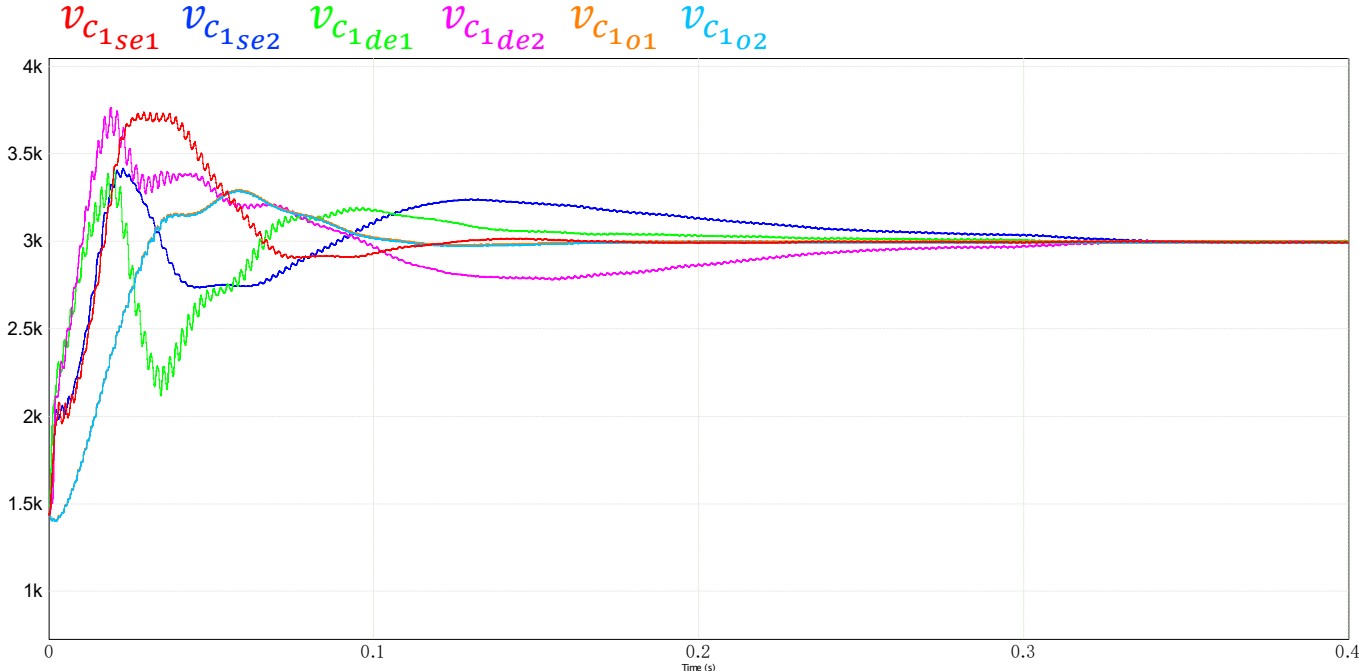

**Figure 21.** Dc-link voltage waveforms for case 6.

On the other hand, variations of the inductor equivalent resistance of up to 10% of its original value have also been simulated, but the results showed no variations in the dc-link voltages, then they are not presented in this work.

### 7. Conclusions

The control strategy presented in this work allows us to balance the energy of the converter arms by means of circulating ac and dc currents. The ac currents circulate only inside the converter not being reflected in the input or output of the topology. The ac current flow through $T_2$-structure is the same that in $T_1$-structure, this allows us to obtain an input current with minimum harmonic content. Moreover, due to the control strategy, the output voltage has also a very low ripple. These characteristics are fundamental in HVdc systems since they are not designed to manage ac variables.

On the other hand, the sensitivity analysis shows that small variations of the converter parameters do not affect the performance of the control system.

The decoupling of the control allows us to reduce the impact of power changes on the output, as seen in the results. The presented topology operating with the proposed control strategy exhibits no common-mode alternating voltage component, which is an advantage and validates its possible use in HVdc or MVdc systems.

**Author Contributions:** Conceptualization, C.P., J.R. and R.P.; methodology, C.P.; software, C.P. and J.R.; validation, C.P. and R.V.; formal analysis, I.A. and W.J.; investigation, I.A. and W.J.; resources, R.V. and W.J.; writing—original draft preparation, C.P. and J.R.; writing—review and editing, R.P.; visualization, W.J.; supervision, J.R.; project administration, R. Villalobos; funding acquisition, R.P. and C.P. All authors have read and agreed to the published version of the manuscript.

**Funding:** This research was funded by the Chilean Agency of Research and Development, grant number "ANID FONDAP 15110019—PUENTE 1522A0006" and "DIUFRO Project support by Universidad de La Frontera grant number DI23-0011".

**Institutional Review Board Statement:** Not applicable.

**Informed Consent Statement:** Not applicable.

**Conflicts of Interest:** The authors declare no conflict of interest.

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
