# Peer review of "Decoupled Control for Double-T Dc-Dc MMC Topology for MT-HVdc/MVdc Grids"

_applsci, doi:10.3390/app13063778_

Round 1

Reviewer 1 Report

This paper presents the decoupled control of a dc-dc MMC based on a double-T topology MT-HVdc/MVdc Grids.

- The author claimed that the converter is made of two T-topologies. However, in line 82, T2 is mentioned as a sub-topology. Is T2 a sub-topology of T1 or T1 and T2 are two sub-topologies? Please correct and stick to a consistent representation.

- In Fig. 2, the term 'series' is spelled incompletely.

- In line 87, the author claimed "each stack (??????1,2 ??????????1,2 and ??????1,2) is formed by N full H-bridges" - does it mean series stack has N full H-bridge, derivation has one and output has one? If so, this increases the number of switches in the circuit and thereby, the switching losses. Please address.

- In line 175, it should be '0.7', not '0,7'. In line 179, it should be '9.1kV' not '9,1kV'. Please check the same for all the places. I can see multiple places where '.' is used as ','.

- Line 256: Figure 10, not Figure 11.

- What kind of modulation is used to control the switches in H-bridge?

- Simulations are an ideal environment. The results would have a stronger impact if supported by experiments (at least in a scaled-down level). This would also support their claim in "Featured Application".

- Please be consistent in using 'double T topology'/'two T-topology' & also '-T topology'/'T-topology'

- Grammatical errors must be addressed.

Reviewer 2 Report

This paper proposes the decoupled control of a dc-dc modular multilevel converter (MMC) based on a double-T topology intended for multi-terminal high voltage direct current (MT- HVdc) power transmission systems or emerging direct current power grids in distribution systems (MVdc). 

1.The reviewer found there were too many grammar errors, please carefully correct them. Currently theses errors make the reviewer cannot understand this paper.

2. Fig.1 is not clear.

3. Captions of Figs. 7-16 should be clearer. 

4.The double-T structure is proposed in this paper to eliminate the input and output filter capacitors, thus decreasing the implementation cost. The reviewer suggest adding the introduction of the conventional structure at the beginning of section 2, where theoretically illustrate the disadvantages of the conventional structure. 5.In addition, could the authors please add the comparative simulation analysis in section 5 for the conventional and proposed structures?

Reviewer 3 Report

The work presents a decoupled control applied to MMC converters in a Double-T configuration. The paper is well written and well structured. However, additional simulation results are needed and it would be interesting to present experimental results.

Below are some specific comments about the work:

1. The authors state that: "The double-T structure allows the operation without input and output filter capacitors decreasing the implementation cost". However, multilevel topologies, in general, do not need input and output filters due to the high number of levels that the converter presents. Also, due to modularity, MMC converters with H-bridge already have this advantage since they can have several modules in series. Thus, I suggest the authors to re-discuss this advantage that they cite as being specific to the double-T structure.

2. The authors state that: "The system has been modeled and simulated results were obtained using MATLAB-PSIM platforms". In the field of power electronics, it is usual for results to be validated on an experimental bench or, at least, on a real-time simulation platform. Therefore, I would like the authors to present a justification for presenting simulation results only. If experimental or real-time results cannot be presented, I ask that the authors provide some technical justification for such absence.

3. Figure 1 has a very small size. I suggest rediagramming the figure in order to leave it with a larger size.

4. The presentation and equation (model) of the converter are duly presented in the paper. congratulations for the work.

5. In the simulation results chapter, a table with the parameters used was expected. For example which inductances (L) were used in each part of the proposed converter? They are all te same? The same goes for capacitances (C) and resistances (R). Also, how were these values defined? What calculations/equations were used? The authors present only capacitance values (1mF) in the "Brief Sensitivity Validation" chapter. As there are several DC-links and several converters, perhaps 1mF is a very large capacitance (which would increase the cost of this converter). Please provide a design metodology in order to justify the choice of these values.

6. Authors use "dot" (.) and "comma" (,) to define decimal numbers. For example, on line 228: "In Fig. 7 at 0.4, 0,8 and 1,2 seconds". Please normalize this by checking the entire paper.

7. Line 256: Replace "Figure 11" with "Figure 10".

8. In the chapter "Brief Sensitivity Validation" the authors present a variation of 1%, 2% and 5% in each capacitance. However, no sensitivity analysis was presented for the inductance variation of the proposed topology. I suggest applying the same methodology to the variation of inductances.

9. Furthermore, the input voltage (Vi) and output voltage (Vo) have not changed during the simulation results. I suggest that the authors increase the sensitivity analysis by inserting a variation in the input voltage (Vi) in order to simulate a disturbance at the input and also by inserting a variation in the output voltage (Vo) simulating a change in the operating setpoint.

Round 2

Reviewer 1 Report

Thank you for addressing the comments.

Reviewer 2 Report

Thanks for the modifications. It can be published in its current form.

Reviewer 3 Report

All my questions are addressed. Congratulations for the work.